# Enhanced Adsorption of Cd on Iron–Organic Associations Formed by Laccase-Mediated Modification: Implications for the Immobilization of Cadmium in Paddy Soil

**DOI:** 10.3390/ijerph192315650

**Published:** 2022-11-25

**Authors:** Weilin Yang, Chunlei Huang, Xiang Wan, Yunyun Zhao, Zhengyu Bao, Wu Xiang

**Affiliations:** 1Faculty of Materials Science and Chemistry, China University of Geosciences, Wuhan 430074, China; 2Zhejiang Institute of Geological Survey, Hangzhou 312000, China; 3State Environmental Protection Key Laboratory of Soil Health and Green Remediation, College of Resources and Environment, Huazhong Agricultural University, Wuhan 430070, China; 4Hubei Geological Survey, Wuhan 430034, China; 5Hubei Key Laboratory of Critical Zone Evolution, School of Earth Sciences, China University of Geosciences, Wuhan 430074, China

**Keywords:** cadmium, iron–organic association, laccase-mediated, adsorption, modification of organic matter

## Abstract

The objectives of this study were to evaluate the cadmium adsorption capacity of iron–organic associations (Fe-OM) formed by laccase-mediated modification and assess the effect of Fe-OM on the immobilization of cadmium in paddy soil. Leaf organic matter (OM) was extracted from Changshan grapefruit leaves, and then dissolved organic matter (Lac-OM) and precipitated organic matter (Lac-P) were obtained by laccase catalytic modification. Different Fe-OM associations were obtained by co-precipitation of Fe with OM, Lac-OM, and Lac-P, respectively, and the adsorption kinetics, adsorption edge, and isothermal adsorption experiments of Cd on Fe-OM were carried out. Based on the in situ generation of Fe-OM, passivation experiments on Cd-contaminated soils with a high geological background were carried out. All types of Fe-OM have a better Cd adsorption capacity than ferrihydrite (FH). The theoretical maximum adsorption capacity of the OM-FH, Lac-OM-FH, and Lac-P-FH were 2.2, 2.53, and 2.98 times higher than that of FH, respectively. The adsorption of Cd on Fe-OM is mainly chemisorption, and the -OH moieties on the Fe-OM surface form an inner-sphere complex with the Cd ions. Lac-OM-FH showed a higher Cd adsorption capacity than OM-FH, which is related to the formation of more oxygen-containing groups in the organic matter modified by laccase. The immobilization effect of Lac-OM-FH on active Cd in soil was also higher than that of OM-FH. The Lac-OM-FH formed by laccase-mediated modification has better Cd adsorption performance, which can effectively inactivate the activity of Cd in paddy soil.

## 1. Introduction

Iron (hydro)oxides are the most abundant metal oxides in soils. Their large surface area, outstanding sorption capacity, and active chemical properties can influence the migration and transformation of heavy metals such as cadmium through sorption and co-precipitation. Iron (hydro) oxides have long been considered to play a crucial role in controlling the fate of heavy metals in soils and sediments [1,2,3]. Organic matter (OM) is an essential ligand in the environment, which is rich in functional groups and heterogeneous binding sites that can effectively bind metals and reduce their bioavailability [4,5]. Iron (hydro)oxides in soils and sediments often combine with OM to form iron–organic complexes or composites (Fe-OM association/coprecipitate/aggregation) [6,7]. The influence of Fe-OM on metal(loid)s is more complex than that of simple iron (hydro)oxides and has therefore received extensive attention [8,9,10]. 

In the vicinity of redox interfaces that are strongly influenced by water levels, such as rice paddies and peatlands, dissolved iron would be oxidized to form a new “reactive iron phase” and forms Fe-OM with organic matter through a variety of actions such as adsorption, co-precipitation, and agglomeration [11,12,13]. The co-composition of OM and iron affects the properties of iron oxides in various aspects, such as the specific surface and surface pore size, surface charge, and roughness of the composite surface [14,15,16]. The presence of organic matter will interfere with the structure of iron oxides and increase their disorder degree and will inhibit the transformation of the iron oxide mineral phase and the formation of a secondary mineral phase catalyzed by Fe (II), making them have a higher point density and larger specific surface area [17,18,19]. In addition, a large number of studies have also shown that iron has a better anti-biological and abiotic decomposition ability after forming co-precipitation with organic matter, and it can exist in the soil environment for a longer time [20,21,22,23]. Fe-OM shows significantly different adsorption properties of heavy metals from pure minerals, and OM will influence the morphology and distribution of heavy metals in the phase transformation of iron (hydro) oxides [10,24,25,26]. Therefore, compared with inorganic minerals, Fe-OM is gradually considered a more critical medium to study the interface behavior of heavy metal ions in soil [27,28,29].

Many studies have confirmed that newly-formed amorphous iron (hydro) oxides preferentially bind phenolics and aromatic organic matter [11,13,30,31,32]. In addition, high molecular weight (MW > 500 Da) organic compounds and highly unsaturated or oxygen-rich organic matter (including polycyclic aromatic hydrocarbons, polyphenols, and carboxylic acid) have a higher affinity for iron (hydro) oxides [33]. Our previous research found that laccase, the most significant type of phenol oxidase in soil, can significantly promote the formation of Fe-OM. The mechanism is that enzymatic polymerization transforms small molecular organic matter into larger molecule DOM with stronger iron affinity and then combines with iron ions to form Fe-OM through coprecipitation [34,35]. In general, organic matter with higher MW diffuse more slowly and have a stronger affinity for binding sites, while more minor MW organic matter tends to be more difficult to immobilize due to its weaker affinity [22,32,36,37]. At present, it is unclear how laccase-mediated Fe-OM affects the environmental behavior of metal(loid)s.

Cadmium is a highly toxic heavy metal and non-essential nutrient element with substantial biological toxicity, irreversibility, accumulation, and non-degradability [38,39], posing a health risk to humans through the food chain [40,41]. Several studies have been reported on the remediation of Cd-contaminated soils based on the interaction between Fe-OM and Cd. Xu et al. used associations of Fe and humus to treat Cd-contaminated soil and passivated 50.7% of active Cd [42]. Recently, plant leaf extracts have been successfully used to synthesize Fe-OM associations. The leaves have a low cost and a wide range of raw materials. In addition, plant leaves contain rich phenolics, flavonoids, carboxylic acids, and amino acids which can be used as both reducing agents and stabilizing agents in the process of Fe-OM [43]. Lin et al. synthesized GION (green synthesized iron oxide nanoparticles) using OM extracted from red-backed laurel (*Excoecaria cochinchinensis*) leaves and iron, which reduced exchangeable Cd and carbonate-bound Cd in the soil by 14.2–83.5% and 18.3–85.8%, respectively [44]. Francy et al. synthesized nanoscale zero-valent iron (nZVI) from *Azadirachta indica* (neem) and *Mentha longifolia* (mint) leaves, which successfully reduced the active Pb and Ni in the soil by 66.1% and 56.8%, respectively [45]. 

The preparation and fates of Fe-OM associations have been deeply studied, and plant extracts can be used for making Fe-OM nanoparticles. Laccase-modified OM can promote the formation of Fe-OM. At present, the fates and environmental effects of laccase-mediated Fe-OM are rarely studied. So, in this study, the organic matter was extracted from the leaves of Huyou (pomelo) and coprecipitated with iron after laccase-mediated modification to prepare Fe-OM associations, and the adsorption experiments of Cd were carried out. The primary purpose of this study is to investigate the adsorption behavior of Cd on Fe-OM formed by laccase-mediated modification to deeply understand its impact on the environmental behavior of heavy metal(Cd) and to provide a green and economical idea for the remediation of Cd-contaminated soil.

## 2. Materials and Methods

### 2.1. Materials and Chemicals

The study area is in Changshan County (28°55′54.15″ N; 118°27′59.87″ E), Quzhou City, Zhejiang Province, where a large amount of leaf waste is produced yearly from the particular agricultural product Huyou (pomelo). The contents of nutrient elements Mg and Ca in Huyou leaves are 0.26% and 3.14%, respectively, while the contents of Fe, Mn, and Cu are 75 mg/kg, 45 mg/kg, and 9.6 mg/kg, respectively [46]. The collected leaves were washed with ultra-pure water, dried, crushed, and passed through a 2 mm sieve.

The soil parent rock of the local paddy soil is the carbonaceous siliceous shale of the Lower Cambrian Hetang Formation, which often develops a black metal layer. Therefore, the content of Cd in the soil seriously exceeds the standard, with it reaching 2.23 mg/kg. Collect the topsoil of paddy soil, remove grassroots, stones and other foreign matter, place the collected soil under natural conditions to dry in the shade, and grind it through a 0.8 mm nylon sieve. 

### 2.2. Preparation and Characterization of Fe-OM

The organic matter was extracted from the leaves. 200 g of the crushed air-dried leaves were put into a 3000 mL beaker, and 2000 mL of 0.1 mol/L NaOH solution was added. Then, it was heated in a water bath at 60 °C and stirred at a speed of 200 r/min for 24 h. The original organic matter solution (OM) was obtained by centrifuging the leaf residues. 1000 mL of OM was taken and the pH was adjusted to 4.7 ± 0.1. After adding 1.0 g laccase (from *Aspergillus oryzae*), it was heated at 50 °C in a water bath with a stirring speed of 200 r/min. Oxygen is introduced into the system throughout the process. After laccase modification treatment for 72 h, the mixture was centrifuged to separate the supernatant and precipitate, which were recorded as laccase-modified organic matter (Lac-OM) and laccase-modified organic matter precipitate (Lac-P), respectively. The content of TOC (total organic carbon) in OM and Lac-OM solutions was measured. The contents of C, H, O, and N in OM were measured by an element analyzer and characterized by Fourier Transform infrared spectroscopy (FT-IR, Nicolet iS50, Thermo Fisher Scientific, Waltham, MA, USA) and Fourier Transform Ion Cyclotron Resonance Mass Spectrometry (FT-ICR-MS, Bruker SolariX™ 2XR, BRUKER, Billerica, MA, USA).

Ferrihydrite (FH) was synthesized according to Schwertmann and Cornell [47]. Briefly, 40.0 g ferric nitrate hydrate (HNO_3_·9H_2_O) was dissolved in 500 mL of deionized water. Then, 330 mL of 1mol/L KOH solution was added and stirred quickly. The last 20 mL was added slowly to keep the system pH between 7–8 to obtain the precipitation of FH. Four different Fe-OM associations (OM1-FH, OM5-FH, Lac-OM1-FH, and Lac-OM5-FH) were obtained by adding OM and Lac-OM solution into the synthesis system of FH, according to the mole ratio of Fe/C 1:1 and 5:1. At the same time, Lac-P-FH was obtained by adding a certain amount of Lac-P into the process of FH synthesis. FH and Fe-OM associations were washed with deionized water until the conductivity of the effluent was <100 μS/cm, freeze-drying and passing a 2 mm sieve. XRD (X-ray diffraction, Bruker D8 Advance, BRUKER, Billerica, MA, USA) was used to determine the crystal morphology and mineral type of Fe-OM. The mineral surface morphology was observed by SEM-EDS (Scanning Electron Microscope-Energy Dispersive Spectroscopy, SU8010, HITACHI, Tokyo, Japan), and the mineral surface composition and element-binding state were determined by XPS (X-ray photoelectron spectroscopy, K-ALPHA, Thermo Scientific, Waltham, MA, USA).

### 2.3. Adsorption Kinetics and Isotherms

The adsorption kinetics experiment was performed in a 250 mL conical flask by taking 0.50 g FH and Fe-OM associations in the conical flask, adding 200 mL of CdCl_2_ solution with a concentration of 50 ppm Cd, using 0.01 M sodium nitrate (NaNO_3_) as a background ion, and maintaining the pH of the solution at 6.0 with 0.5 mM MES/MOPS buffer system. The conical flask was placed on a reciprocating shaker at 250 r/min, and 1.0 mL of the solution was taken at 1 min, 5 min, 10 min, 20 min, 30 min, 45 min, 60 min, 120 min, 240 min and 360 min, centrifuged at 10,000 r/min, and filtered through a 0.45 μm membrane. The supernatant was diluted with 1% secondary purified HNO_3_ at a specific multiple. The Cd concentration was measured by AAS (Flame Atomic Absorption Spectroscopy, NovAA 800, Analytik Jena, Jena, Germany). 

Adsorption edge experiments were performed at four different pH levels (4, 5.5, 7, and 8.5) and three different concentrations of background ion solutions (0.1 M, 0.01 M, and 0.002 M sodium nitrate solution), using an MES/MOPS buffer system to adjust the pH 4 and 5.5 and tris-HCl buffer system for pH 7 and 8.5. 10mg FH and Fe-OM associations and 10mL CdCl_2_ solution were added into a 15 mL centrifuge tube and then shaken at 200r/min for 24 h at room temperature. The centrifuge tube was centrifuged at 3000 r/min for 10 min and filtered through a 0.45 μm membrane. 

The concentrations of Cd solution used for isothermal adsorption were 5 ppm, 10 ppm, 20 ppm, 40 ppm, 70 ppm, 100 ppm, 150 ppm, and 200 ppm. 0.01 M NaNO_3_ was used as the background ion, and 0.5 mM MES/MOPS buffer system was used to adjust the pH to 6.5. 10 mg FH and Fe-OM associations and 10 mL CdCl_2_ solution were added into a 15 mL centrifuge tube and then shaken at 200 r/min for 24 h at room temperature. The centrifuge tube was centrifuged at 3000 r/min for 10 min and filtered through a 0.45 μm membrane. 

Take 1.00 g of FH, Lac-OM1-FH and OM1-FH into a 250 mL conical flask, then add 200 mL of Cd solution with a concentration of 1000 ppm and pH 6.0. After shaking on a reciprocating oscillator at a speed of 250 r/min for 4 h, the supernatant was removed by centrifuge. The remaining iron minerals adsorbed with Cd were washed with 10mL ultra-pure water, and the Cd-adsorbed FH and Fe-OM associations were obtained after freeze-drying. XPS and SEM-EDS were used to characterize the surface morphology and element-binding state of FH and Fe-OM associations. 

### 2.4. Adsorption Kinetics and Isotherm Models and Parameters 

To gain a deeper insight into the adsorption kinetics, five models (Elovich, two-constant equation, intra-particle diffusion model, pseudo-first order kinetics equation, and pseudo-second order kinetics equation) were employed to fit the adsorption kinetics of Cd onto FH and Fe-OM associations in this study. 

The Elovich kinetic model is a semi-empirical model used to describe the chemical adsorption behavior of adsorbate on heterogeneous surfaces and is given by Equation (1).
Qt = a + bInt (1)

The two-constant equation is an empirical model which is mainly applicable to the kinetic process with more complex reactions and is expressed by Equation (2).
InQt = a + bInt(2)

The intra-particle diffusion model is mainly used to determine the control steps of the adsorption process rate and is given by Equation (3).
Qt = Kt^1/2^(3)

Based on the assumption that the rate of change of adsorbed solute with time is proportional to the difference in equilibrium adsorption capacity and the adsorbed amount, the pseudo-first order kinetics model can be expressed by Equations (4) and (5).
dQ/dt = K(Qe − Qt)(4)

In linear form: In(1 − Qt/Qe) = −Kt(5)

The pseudo second-order kinetics model is based on the assumption that the rate-limiting step involves chemisorption which involved covalent forces through sharing or exchange of electrons between the adsorbent and adsorbate and is described by Equations (6) and (7).
dQ/dt = K(Qe − Qt)^2^(6)

In linear form: t/Qt = 1/(KQe^2^) + t/Qe(7)
where t (min) represents the adsorption time (min), Qt (mg/g) represents the amount of Cd^2+^ adsorbed at time t, Qe (mg/g) represents the amount of Cd^2+^ adsorbed at equilibrium (mg/g), and a, b, K represent various corresponding adsorption kinetic constants. 

The Langmuir model and Freundlich isotherm model were used to fit the isothermal adsorption of Cd onto FH and Fe-OM associations.

The Langmuir model assumes that monomolecular layer adsorption occurs on specific heterogeneous surfaces where all adsorption sites are identical and energetically equivalent as Equations (8) and (9).
Qe = Q_m_Ce/(1/K + Ce)(8)

In linear form: Ce/Qe = 1/KQ_m_ + Ce/Q_m_(9)

The Freundlich isotherm model is commonly employed for the adsorption of a reversible heterogeneous surface. The Freundlich isotherm equation is represented as Equations (10) and (11).
Qe = KCe^1/n^(10)

In linear form: InQe = InK + nInCe(11)
where Qe (mg/g) is the amount of Cd^2+^ adsorbed at equilibrium, Ce (mg/L) is the concentration of Cd^2+^ in the remaining solution at equilibrium (mg/L), Q_m_ (mg/g) is the theoretical maximum adsorption amount, and K and n are the corresponding isothermal adsorption constants.

### 2.5. Soil Passivation Experiment

In order to evaluate the remediation prospects of Fe-OM associations on Cd-contaminated paddy soil, soil simulation incubation experiments were conducted. The incubation experiments were conducted in a 2500 mL polypropylene bucket. 1.5 kg of contaminated soil was taken into the bucket, Lac-OM1-FH and OM1-FH were added at 1% of soil mass and stirred evenly, and deionized water was added to keep the moisture content at 60% at room temperature. After 30 days of incubation, column samples of cultured soil were collected using 25 mm diameter PVC tubes. After freeze-drying, the active Cd, the Cd species and pH values in the soil were measured. The active Cd content was determined by the DTPA method and the Cd species was determined by the Tessier sequential chemical extraction procedure.

### 2.6. Statistical Analysis

The FT-ICR-MS measurement data of organic matter and laccase modified organic matter was analyzed with Composer 15.6 (Sierra Analytics, Modesto, CA, USA). 

All of the model parameters and the correlation coefficient (R^2^) were evaluated by non-linear regression using OriginPro 2022 SR1 software. Expect figure of SEM-EDS, all of the other figures were drawn with OriginPro 2022.

## 3. Results and Discussion

### 3.1. Characterization of Materials

The molecular weight (MW), element composition, and other information of each organic matter molecule in the extracted organic matter were obtained by FT-ICR-MS. According to the MW of organic matter molecules in OM and Lac-OM, they can be divided into six different MW intervals. The number of molecules in each interval is counted and drawn in Figure 1. It can be seen that in OM and Lac-OM, the number of molecules with MW in 300–500 Da is the largest, accounting for 54.6% and 58.8%, respectively, followed by 200–300 Da and 500–600 Da intervals, and the smallest number of molecules in the <100 Da and >700 Da interval. 

According to the H/C molar ratio and the modified aromatic index (Almod) of organic matter molecules and the experience of Merder et al. [48], the extracted OM could be divided into five types: PAHs (polycyclic aromatic hydrocarbons), aliphatic compounds, polyphenols, carbohydrates, and highly unsaturated lignin and phenols. The molecular numbers of these five types of organic matter in OM and Lac-OM are shown in Figure 1. The highest number of molecules in both OM and Lac-OM are highly unsaturated lignin and phenols, followed by aliphatic compounds, and the lowest are PAHs and carbohydrates. The organic composition of leaves is mainly composed of cellulose, hemicellulose, and lignin. Cellulose is a macromolecule consisting of multiple glucose monomers linked by β-1,4-glycosidic bonds. Hemicellulose is a polysaccharide composed of structural units such as five-carbon monosaccharides and six-carbon monosaccharides. Lignin is a kind of complex natural organic macromolecular compound with an aromatic ring structure, which exists in the cell wall of higher plants. It is mainly composed of three basic units, syringyl, guaiacyl, and p-hydroxybenzyl, which are connected to form a complex macromolecular 3D structure through various C-O-C (carbon-oxyether) bonds and C-C (carbon-carbon bridge) bonds. Lignin is connected with hemicellulose through a covalent bond and filled between cellulose. The source of OM in this study is the alkaline decomposition of leaves. The alkali solution can destroy the lignocellulose structure, weaken the hydrogen bond between cellulose and hemicellulose, and saponify the lipid bond between hemicellulose and lignin, separating lignin, cellulose, and hemicellulose, thus eluting lignin [49,50,51]. Therefore, the OM extracted from leaves in this experiment contains a lot of lignin.

FT-ICR-MS can characterize the number of C, H, and O atoms of different organic molecules. With the O/C molar ratio of each type of organic matter as the abscissa and H/C molar ratio as the ordinate, different kinds of organic matter molecules in OM and Lac-OM are mapped into the VK (Van Krevelen) diagram (Figure 2). The blue dots in the figure represent OM-specific organic matter, that is, organic matter decomposed or precipitated after laccase modification; the red dot represents the unique organic matter in Lac-OM, that is, the newly formed soluble organic matter after laccase modification; the grey dot represents the organic matter shared by OM and Lac-OM.

Figure 1 shows that after OM was modified by laccase and produced precipitation, the remaining Lac-OM had a smaller number of molecules than OM in each molecular weight segment and in different organic matter species. Combined with Figure 2, it proves that laccase did modify OM to some extent. Laccase utilizes oxygen as an electron acceptor to catalyze the modification of organic compounds such as phenolics, aromatics, amines, and aliphatic groups to form the corresponding reactive radicals or quinone intermediates, and the oxidized radicals can participate in other reactions to form by-products or generate polymers by coupling [35,52]. Therefore, the catalytic modification process of laccase for complex organic matter includes degradation and polymerization reactions. In the present experiments, the treatment of OM by laccase resulted in the degradation of numerous organic matter molecules on the one hand, resulting in a decrease in the number of organic matter molecules and an overall decrease in MW. On the other hand, some small organic molecules are polymerized by catalysis, generating insoluble precipitates of large organic matter (Lac-P). After measurement, the TOC content in OM and Lac-OM was 3.09% and 2.21%, respectively.

Figure 3 shows the FT-IR spectra of OM and Lac-OM. A prominent adsorption peak between 3600–3000 cm^−1^, indicating the presence of both phenolic and alcoholic O-H groups and N-H proteinaceous moieties. The band at 1524.63 cm^−1^ may be due to C-C stretching vibrations in lignin aromatic moieties [53,54]. The peak at 1400 cm^−1^ is generated by the C-O stretching of phenolic O-H and C-H deformation of CH_2_ and CH_3_ groups [55]. The signal at 1239.22 cm^−1^ can be attributed to the C-O stretching vibration and O-H deformation vibration of carboxylic acid functional groups. Peaks at 2926.62 cm^−1^ and 2844.52 cm^−1^ can be assigned to C-H stretch vibration in aliphatic methylene, and the peak at 1634.74 cm^−1^ can be assigned to O-H bend vibration [56]. The bands at 1033.58 cm^−1^ represent the O-H stretch vibration. The three peaks with the highest peak intensities (3279.02 cm^−1^, 1634.74 cm^−1^, and 1033.58 cm^−1^) represent -OH in both OM and Lac-OM, indicating that -OH functional groups exist in large amounts. By comparing the FT-IR spectra of OM and Lac-OM, it can be found that both of them contain basically the same signal peaks, representing that the composition of OM functional groups is basically the same before and after laccase modification treatment, and the catalytic modification of laccase does not cause OM to produce new functional groups.

### 3.2. Adsorption Edge

Adsorption edge experiments investigated the effects of ionic strength and pH on the adsorption behavior of Cd onto the surface of FH and Fe-OM associations (Figure 4). The results showed that the variation of ionic strength did not have a significant effect on the adsorption of Cd. The effect of ionic strength on Cd adsorption can be used to distinguish the types of complexes formed after Cd adsorption. When the adsorption process is insensitive to ionic strength or the adsorption increases with the increase of ionic strength, it indicates that Cd enters the inner layer of the double electric layer, which is directly united with the hydroxyl function group on the mineral surface and forms an inner-sphere complex through the inner layer adsorption. However, when the adsorption decreases with the increase of ionic strength, it indicates that Cd and the electrolyte in the solution have competitive adsorption on the mineral surface, forming ion pairs through electrostatic interactions with electrocharged groups on the mineral surface and forming the outer-sphere complex [57,58]. Therefore, Cd is directly bound to the -OH group on the surface of Fe-OM to form an inner-sphere complex. 

pH value significantly affects Cd adsorption on FH and Fe OM associations. The adsorption of Cd by FH and Fe-OM showed a sharp increase after the pH exceeded 5.5. When the pH exceeded 7.0, the increase in Cd adsorption by FH, OM5-FH, and Lac-OM5-FH all showed a pronounced tendency to slow down. While the adsorption of Cd by Lac-P-FH, OM1-FH, and Lac-OM1-FH continued to increase with the increase in pH. The isoelectric point (i.e.p) of FH ore is between 8.0 and 8.7. The rise in pH causes the mineral surface to be shielded by negative ions, which strengthens the electrostatic attraction with Cd, and finally results in the enhanced adsorption effect on Cd. Both FH and Fe-OM have a large number of oxygen-containing functional groups such as -COOH and -OH, and surface hydroxyl groups can be protonated and deprotonated under different pH conditions. The Fe-OM surface is negatively charged, and with the constant increase in pH value, the surface will be deprotonated to form new negative surface sites, which is conducive to the solid binding with heavy metal cations [59,60,61]. The content of OM in Lac-P-FH, OM1-FH, and Lac-OM1-FH is higher than that in FH, OM5-FH, and Lac-OM5-FH, which may be attributed to their increased oxygen-containing functional groups. After the pH exceeds 7, the three Fe-OM containing more OM (Lac-P-FH, OM1-FH, and Lac-OM1-FH) still do not reach the adsorption equilibrium, maintaining a good adsorption potential, while the adsorption sites on the surface of the FH and Fe-OM with less OM are gradually consumed, resulting in their inability to adsorb more Cd.

### 3.3. Adsorption Kinetics

As seen from Figure 5, the adsorption kinetics of both FH and Fe-OM exhibit similar characteristics of the two stages. The first stage is a rapid adsorption process with a significant slope of the adsorption curve and a rapid increase in the adsorption amount until 30 min, after which the adsorption rate decreases dramatically. The adsorption of Cd by FH, OM5-FH, and Lac-OM5-FH had basically equilibrated at 120 min, reaching 97.4%, 96.0%, and 94.6% of the equilibrium adsorption amount, respectively. In contrast, Lac-P-FH, OM1-FH, and Lac-OM1-FH only reached about 87% of the equilibrium adsorption amount at 120 min, then the adsorption rate slowed down and finally reached the adsorption equilibrium after 360 min. The adsorption kinetic characteristics of these six adsorbents all conformed to the ideal adsorption curve, which was mainly caused by the fact that there was a time gap in the adsorption of cadmium by adsorption sites with different energies on the adsorbent surface. In the early stage, the high-energy adsorption sites adsorb cadmium ions first. Then, the low-energy sites adsorb the remaining cadmium ions as the adsorption proceeds [62], so there are two processes of initial fast adsorption and late slow adsorption. 

The kinetic behaviors of Cd adsorption by FH and Fe-OM associations were fitted with the Elovich equation, two-constant equation, pseudo-first order kinetics equation, pseudo-second order kinetics equation, and intra-particle diffusion model. The fitting results are shown in Figure 6, and the fitted parameters are listed in Table 1. The pseudo-first order kinetics equations fit the Cd adsorption process worse than the pseudo-secondary kinetics equations, especially for FH, OM5-FH, and Lac-OM5-FH, the correlation coefficient values (R^2^) for first-order rate expression were only 0.795, 0.738, and 0.871, respectively, while the R^2^ for second-order rate expression was in all cases greater than 0.997. Wang and Guo summarized the conditions for the applicability of pseudo-first order kinetics equation: a very high initial concentration of adsorbent, only the initial stage of adsorption was calculated, and few active sites of the adsorbent [63]. The initial concentration of Cd^2+^ in the present experiment was only 50 mg/L, and the whole adsorption process was calculated, so the pseudo-first order kinetics equation could only fit the preliminary behavior of the adsorption reaction best, and wit as not applicable to fit the whole adsorption process. In contrast to the pseudo-first order kinetics equation, the pseudo-secondary kinetics equation can better fit the adsorption behavior when the initial concentration of adsorbent is low. The adsorption is a complex non-homogeneous diffusion process subject to multiple kinetic mechanisms. The initial stage of Cd(II) adsorption is mainly external liquid film diffusion, followed by the intra-particle diffusion process until the adsorption reaches equilibrium. The adsorption process is dominated by chemisorption. Both the Elovich equation and the two-constant equation fit the adsorption process better, and the R^2^ of the fit is greater than 0.97 for all four Fe-OM associations except for FH and OM5-FH. Elovich is an empirical model for the adsorption processes on non-homogeneous surfaces [64]. The Elovich model and the fitting junction of the double constant equation show that the adsorption of mineral Cd by FH and Fe-OM is a complex heterogeneous diffusion process which is affected by various mechanisms, further confirming that the adsorption process is mainly chemical adsorption. The fitting results of Elovich and two-constant equations show that the adsorption of Cd onto FH and Fe-OM associations is a sophisticated non-homogeneous diffusion process, subject to multiple mechanisms, further confirming that the adsorption process is mainly chemisorption.

### 3.4. Adsorption Isotherms

Figure 6 shows the isothermal adsorption curves of different initial Cd concentrations adsorbed by FH, Fe-OM associations, and Lac-P at 25 °C and pH = 6.5. When the initial Cd concentration was less than 70 mg/L, the adsorption of Cd by these adsorbents increased rapidly. After the initial Cd solution concentration exceeded 70 mg/L, the growth rate of the adsorption of Cd decreased significantly and gradually reached the adsorption equilibrium. The amount of Cd adsorbed by Lac-P, Lac-P-FH, OM5-FH, and Lac-OM5-FH remained increasing, and the adsorption equilibrium was gradually reached until the initial Cd concentration reached 150 mg/L.

The Langmuir isothermal adsorption model is suitable for the adsorption process of a homogeneous surface single-molecular layer liquid phase system, while the Freundlich equation is mainly used for the adsorption process between multiphase surfaces. To determine the type of Cd adsorption by Fe-OM associations, the experimental data obtained at different concentrations were plotted in the linear form of the Langmuir and Freundlich adsorption isotherms (Figure 6), and the parameters of the fitted equations, as well as the fitted correlation coefficients, are listed in Table 1. Both Langmuir and Freundlich equations can fit the isothermal adsorption process of FH and Fe-OM on Cd well, and the R^2^ of adsorption fitting exceeds 0.9753 and 0.9627. The fitting result of the Langmuir equation is slightly better than that of the Freundlich equation, which indicates that the adsorption of Cd by FH and Fe OM belongs to single-layer adsorption. Since the multi-layer adsorption mainly occurs in the physical adsorption process, it further confirms that the adsorption of Cd by FH and Fe-OM belongs to chemical adsorption.

It can be seen from the adsorption edge experiment (Figure 4), kinetic curve (Figure 5) and isothermal adsorption curve (Figure 6) that under the same adsorption conditions, the adsorption capacity of FH for Cd is less than that of Fe-OM associations. The theoretical maximum adsorption capacity (Q_m_) of Cd onto FH under laboratory conditions was calculated by the Langmuir equation to be 39.53 mg/g (Table 1), while the Q_m_ of Fe-OM ranged from 59.52–117.65 mg/g, indicating that the Fe-OM significantly increased the adsorption capacity for Cd. In descending order, the Q_m_ of these five Fe-OM associations was Lac-P-FH, Lac-OM1-FH, OM1-FH, Lac-OM5-FH, OM5-FH. The following points can be found by comparison:Lac-P-FH has the best adsorption capacity.Fe-OM with high organic matter content has better adsorption performance for Cd.Lac-OM-FH has better adsorption capacity than OM-FH.

The isothermal adsorption experiments of Lac-P showed a higher Qm (68.97 mg/g) than that of FH but it was less than that of Lac-P-FH, indicating that the co-precipitation of Lac-P and FH greatly enhances the adsorption of Cd. Interactions of organic matter and iron hydroxide can occur through two common processes: OM is directly adsorbed on pristine FH or FH is precipitated in the presence of OM [7,60]. 

OM-adsorbed iron oxide would not affect the crystalline state and disorder internal to the iron oxide, whereas the OM-coprecipitated iron oxide increases the disorder and electromagnetism of the internal structure [65,66]. The Q_m_ of the synthesized Lac-P-FH in the present study for Cd is 200% and 70% more than that of FH and Lac-P, implying that Lac-P is involved in the formation of FH, which eventually forms co-precipitated Fe-OM associations. On the one hand, Lac-P disturbs the crystalline state and disorder of FH and increases the specific surface area; on the other hand, the large number of oxygen-containing groups in Lac-P provides more adsorption sites for the Lac-P-FH. The interaction of the Lac-P and FH contributed to the increase of adsorption site density on the surface of Lac-P-FH, which finally manifested itself in the significantly enhanced Cd adsorption capacity. 

Many studies have explored the characterization of Fe-OM associations at different Fe/C ratios and found that with increasing C content in the associations, Fe-OM has smaller particles, more adsorption space, and higher adsorption potential [61,67,68,69]. More organic matter can provide more oxygen-containing functional groups on the one hand, thus increasing the adsorption sites for Cd; on the other hand, it will increase the disorder in the formation of Fe-OM and form smaller particles, so that the adsorption space of Fe-OM is larger and the density of surface adsorption sites increases.

The Q_m_ of Cd by Lac-OM1-FH and Lac-OM5-FH was 15.0% and 15.07% greater than that by OM1-FH and OM5-FH, respectively. The adsorption of Cd by the co-precipitation of Fe and Lac-OM was significantly higher than that by the co-precipitation of Fe and OM, indicating that the OM modified by laccase has a higher enhancement in the adsorption performance of Fe-OM. Laccase is able to catalyze the modification of a wide range of aromatic compounds (especially phenolics), using the redox capacity specific to its active center T1-Cu to generate reactive radical intermediates that lead to various non-enzymatic secondary reactions such as polymerization, hydroxylation, and disproportionation [70]. The modification of OM by laccase can break covalent bonds to release monomeric substances or perform ring-opening reactions that introduce additional oxygen-containing groups [71,72]. Therefore, the formation of co-precipitation between iron and laccase-modified OM may result in the surface of Fe-OM containing more oxygen-containing functional groups, increasing the density of surface adsorption sites and improving the adsorption capacity of Fe-OM for Cd.

### 3.5. Adsorption Mechanism

In order to further study the adsorption mechanism of Cd on Fe-OM associations, a series of characterizations of FH, La-P-FH, OM1-FH, and Lac-OM-FH after Cd adsorption were carried out by XRD, XPS, and SEM-EDS. FH is an amorphous mineral with no clear peaks in the XRD pattern (Figure 7), and it is generally considered a two-line ferrihydrite. The XRD patterns of Fe-OM associations were similar to those of FH, and the co-precipitation of OM with FH did not cause a transformation of the iron mineral phase, which remained as a two-line ferrihydrite. Cd-adsorbed FH and Fe-OM showed three prominent peaks characteristic of Cd minerals, and it was tentatively judged that these three peaks corresponded to CdCO_3_ (23.67 degrees), Cd(OH)_2_ (29.44 degrees), and CdO_2_ (33.85 degrees) through referring to the PDF card, thus indicating that Cd was indeed bound to FH and Fe-OM with each other and possibly formed independent mineral precipitation. Among the four Cd adsorbents, the peak intensity of Cd in Lac-P-FH-Cd is the highest, followed by Lac-OM-FH-Cd, and OM1-FH-Cd and FH-Cd being the lowest, indicating that Lac-P-FH-Cd and Lac-OM-FH have the strongest adsorption capacity for Cd, which is consistent with the theoretical maximum adsorption capacity obtained by Langmuir isothermal adsorption.

SEM-EDS images (Figure 8) show that the FH and Fe-OM still have no apparent crystalline morphology after the adsorption of Cd. The EDS mapping results show that the Fe and Cd signals correlate well with each other, indicating that the adsorbed Cd is uniformly distributed in the associations.

In general, the adsorption of heavy metals mainly involves two adsorption mechanisms, chemical bonding and physical adsorption [73]. From the above discussion of the adsorption edge, it was found that the adsorption of Cd is not sensitive to the response of background ion concentration, which implies that electrostatic attraction is not the primary mechanism for the adsorption of Cd ions by FH and Fe-OM. The adsorption isotherms and kinetics indicate that the adsorption process is controlled by a chemical mechanism with a homogeneous adsorption activity center. 

XPS analysis was performed on the samples before and after adsorption to further elucidate the adsorption mechanism. Figure 9 shows the total XPS spectra, Cd3d spectra, C1s spectra, Fe2p spectra, and O1s spectra of each adsorbent before and after adsorption. As can be seen from Figure 9a, the four adsorbents show prominent Cd3d peaks after the adsorption of Cd, which is consistent with the XRD and SEM-EDS results, demonstrating that Cd is indeed adsorbed on the surface of FH and Fe-OM associations. By comparing the peak intensities, it was found that the Cd3d peak intensities of Lac-P-FH and Lac-OM1-FH were the highest, while the Cd3d peak intensities on FH were the lowest, which also confirmed the difference in the adsorption performance of each adsorbent. Lac-P-FH and Lac-OM1-FH were significantly stronger than FH. 

The C1s high-resolution spectra before and after adsorption show that no new C peaks appear after the adsorption of Cd, but the relative contents of the three sub-peaks change and the binding energies of the C1s sub-peaks at 288.09 eV (C=O) and 286.23 eV (C-O) appear to be elevated. This indicates that the adsorption of Cd has a significant effect on the C-O and C=O bonds in the FH and Fe-OM composition. Analysis of the high-resolution spectra of Cd3d after adsorption of Cd by the four adsorbents showed that the bimodal binding energies of Cd3d were consistent, both appearing at 405.4 eV and 412.3 eV, and the percentage composition of the bimodal peaks was also consistent, with the peak at 405.4 eV accounting for 52.14% and the peak at 405.4 eV accounting for 47.86%. 

The high-resolution spectra of Fe2p3 showed that Fe had three sets of double peaks in all four minerals, and no new peaks appeared or changes in binding energy occurred before and after the adsorption of Cd, demonstrating that the morphology of the adsorbent did not change during the adsorption process. The proportion of the Fe2p3 peak area in FH and Fe-OM did not change significantly before and after Cd adsorption, suggesting that the adsorption of Cd by these four adsorbents does not depend on Fe, which is consistent with the conclusion obtained by Tian et al. [74] in studying the adsorption of Cd on the complex of pyrite and biochar.

From the high-resolution XPS patterns of O1s in Figure 9e, it can be seen that the peak area and peak intensity represented by O-H in FH are significantly lower than those of the three Fe-OM associations, indicating that the participation of organic matter in the formation process of FH does introduce a large number of oxygen-containing groups for Fe-OM. The intensities of the O1s peaks of all four adsorbents showed a significant decrease after the adsorption. However, the peak areas and corresponding binding energies of the different adsorbents showed different changes after the adsorption. The peak area corresponding to O-H in FH (532.18 eV) showed a certain degree of increase after the adsorption of Cd, while the peak area corresponding to O-H of the three Fe-OM associations showed a significant decrease after the adsorption of Cd. The decrease in O-H peak intensity and peak area is attributed to the exchange of H^+^ in -COOH and -OH with Cd^2+^ during the adsorption process, allowing Cd^2+^ to be adsorbed by chemical bonding [75,76]. Since Fe-OM has a large amount of O-H, it exhibits a much higher adsorption performance than FH. Except for FH, the two peaks at 529.9 eV and 531.3 eV of the other three Fe-OM did not show significant peak drift after the adsorption of Cd, indicating that O-C and O-Fe were not involved in the adsorption process of Cd.

### 3.6. Immobilization of Cd in Contaminated Paddy Soil

After the preliminary investigation, the content of DTPA-Cd in contaminated paddy soil was 0.91 mg/kg, the content of total Cd was 2.23 mg/kg, and the pH was 5.9. After 30 days of passivation incubation, the soil pH values of OM1-FH and Lac-OM1-FH treatment groups were 5.91 and 5.95, respectively. The DTPA-Cd content was reduced by 0.47 and 0.59 mg/kg, passivated 51.6% and 64.8% of the available Cd in the paddy soil, and the passivation effect of Lac-OM1-FH on soil DTPA-Cd was 25.6% higher than that of OM1-FH. The concentration of each species of Cd in the soil after incubation was measured using the Tessier method (Figure 10).

Figure 10 showed that most exchangeable Cd was converted to iron–manganese oxide-bound Cd after treatment, indicating that the synthesized Fe-OM affected the distribution of Cd in soil. In the adsorption experiment, the adsorption capacity of Cd by Lac-OM1-FH is 15% higher than that by OM1-FH. In the soil system, the immobilization ability of Lac-OM1-FH to active Cd is 25.6% higher than that of OM1-FH. The reason why the fixation effect of Lac-OM1-FH to Cd in the soil is higher than that in aqueous solution may be due to the following aspects: the soil solution contains weak acids such as carbonic acid, silicic acid, phosphoric acid, humic acid, and other organic acids and their salts, which constitute an excellent buffering system. The soil buffer system can continuously provide -OH, promote the deprotonation of oxygen-containing functional groups on the surface of Fe-OM, and fully activate the adsorption sites on the surface of Fe-OM. In contrast, the aqueous system cannot provide enough -OH. Therefore, the adsorption capacity of Cd by OM1-FH and Lac-OM-FH was somewhat inhibited. Another possible reason involves the dissolution-recrystallization process of Fe-OM in soil. Fe-OM is reduced and dissolved under the reduction conditions and microbial action to release Lac-OM, OM, and Fe, and the affinity between Lac-OM and Fe is more robust than that of OM. Therefore, Lac-OM can crystallize with Fe better and faster to generate Fe-OM. In this process, active Cd is fixed through adsorption and co-precipitation, which may form the ternary precipitation of Fe, OM, and Cd, so that the fixation capacity of Cd in the soil is more vital than that in aqueous solution. Due to the complexity of microbial-mediated soil processes, the mechanisms need to be further studied. 

## 4. Conclusions

In soils and sediments, the associations of Fe-OM control the geochemical activity and bioavailability of heavy metals such as Cd. Co-precipitation of OM with Fe does not change the crystalline form of the mineral and remains amorphous. The results of Cd adsorption experiments show that Fe-OM associations are much higher than inorganic iron minerals (FH) in adsorption rate and amount. The higher the OM content in the Fe-OM associations, the stronger the adsorption effect on Cd, and the better the Fe-OM adsorption effect formed by OM after laccase modification. The adsorption mechanism of Fe-OM on Cd is mainly chemisorption. Cd enters the inner layer of the double electric layer of the associations molecule and directly bonds with -OH to form an inner-layer complex. The OM modified by laccase has more oxygen-containing functional groups, including hydroxyl groups, which can provide more adsorption sites after co-precipitating with Fe. Fe-OM associations formed by laccase-mediated modification as passivators could reduce the available Cd in the contaminated paddy soil, indicating that laccase-mediated co-precipitation of OM and iron has excellent adsorption properties on Cd and has great potential for practical applications in soil remediation.

## Figures and Tables

**Figure 1 ijerph-19-15650-f001:**
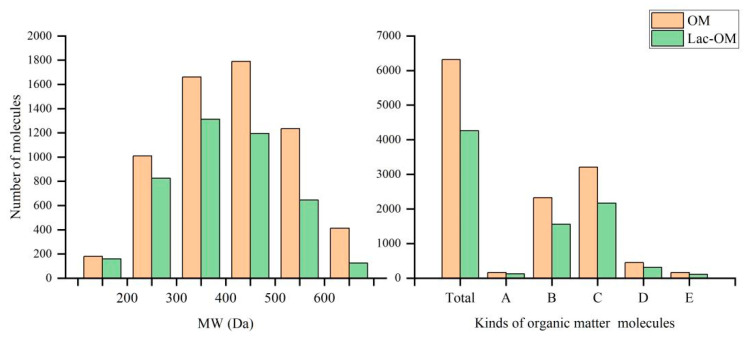
Number of organic matter molecules in different molecular weight intervals and organic matter types (A, polycyclic aromatic hydrocarbons; B, aliphatics; C, lignin, phenolic, highly unsaturated; D, polyphenols; E, carbohydrates).

**Figure 2 ijerph-19-15650-f002:**
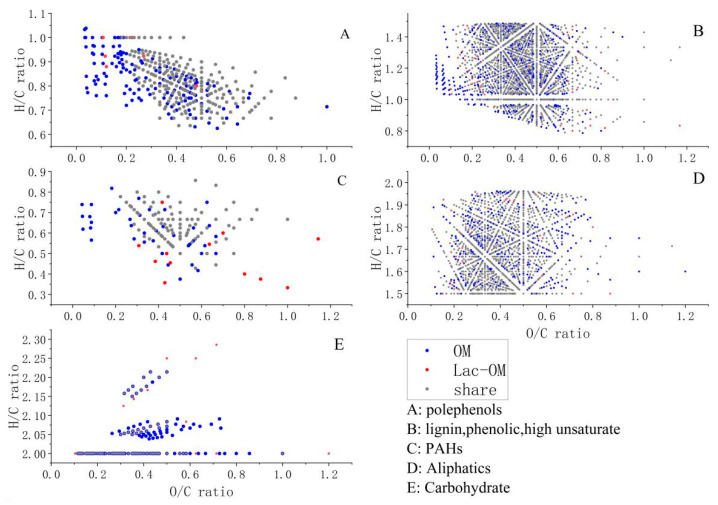
Van Krevelen plot of organic matter derived from leaf and laccase-modified organic matter.

**Figure 3 ijerph-19-15650-f003:**
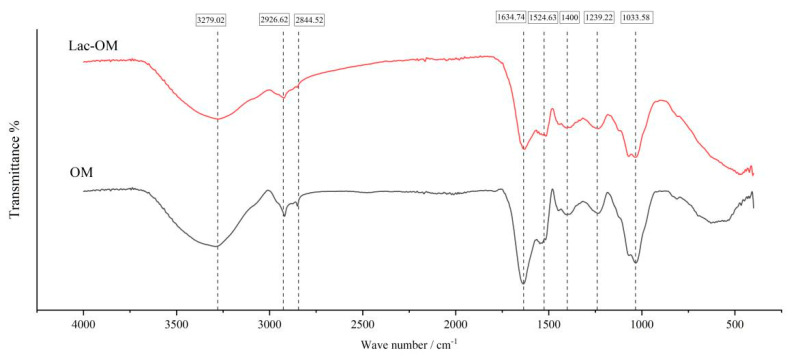
FTIR of organic matter derived from leaf and laccase-modified organic matter.

**Figure 4 ijerph-19-15650-f004:**
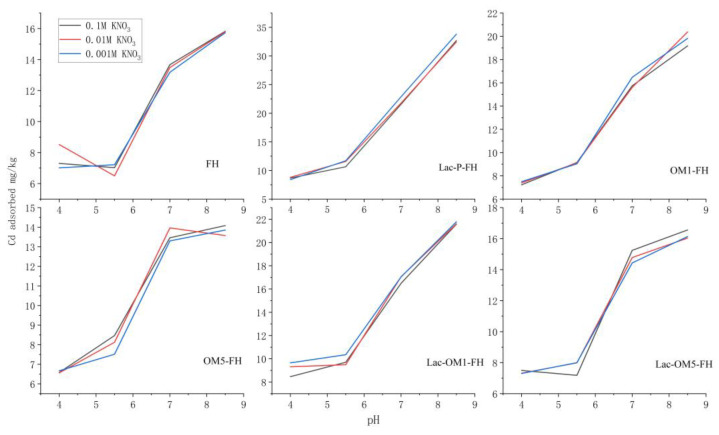
Adsorption edge patterns of Cd onto ferrihydrite (FH) and five iron–organic matter associations (Fe-OM).

**Figure 5 ijerph-19-15650-f005:**
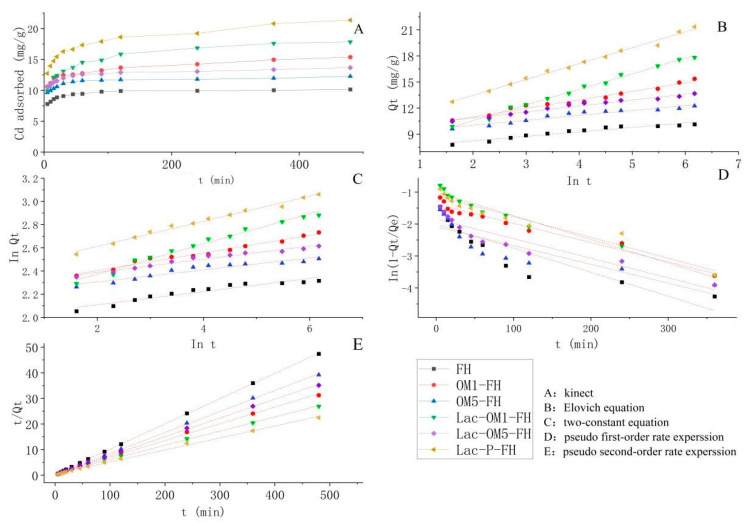
Adsorption kinetics plots of cadmium on ferrihydrite and iron–organic matter associations.

**Figure 6 ijerph-19-15650-f006:**
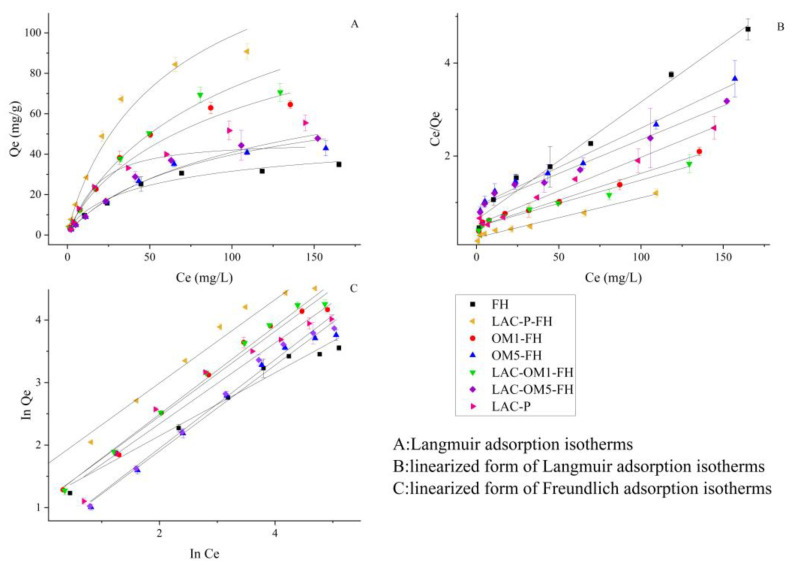
Equilibrium isotherm model analyses with the Freundlich and Langumuir models.

**Figure 7 ijerph-19-15650-f007:**
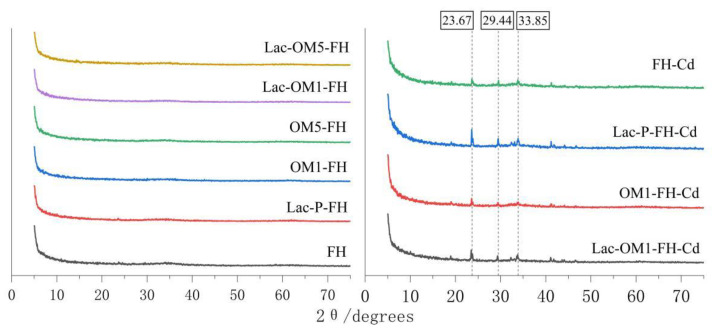
XRD patterns of FH and Fe-OM associations before and after cadmium adsorption.

**Figure 8 ijerph-19-15650-f008:**
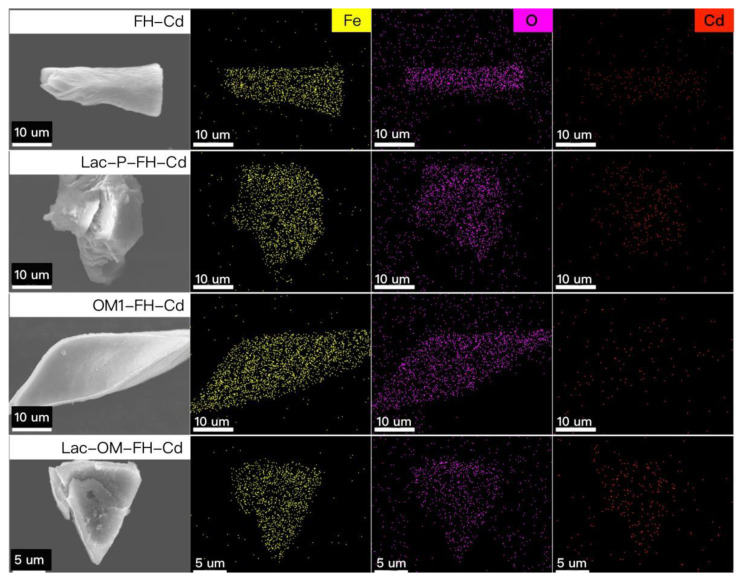
SEM-EDS patterns of FH and Fe-OM after cadmium adsorption.

**Figure 9 ijerph-19-15650-f009:**
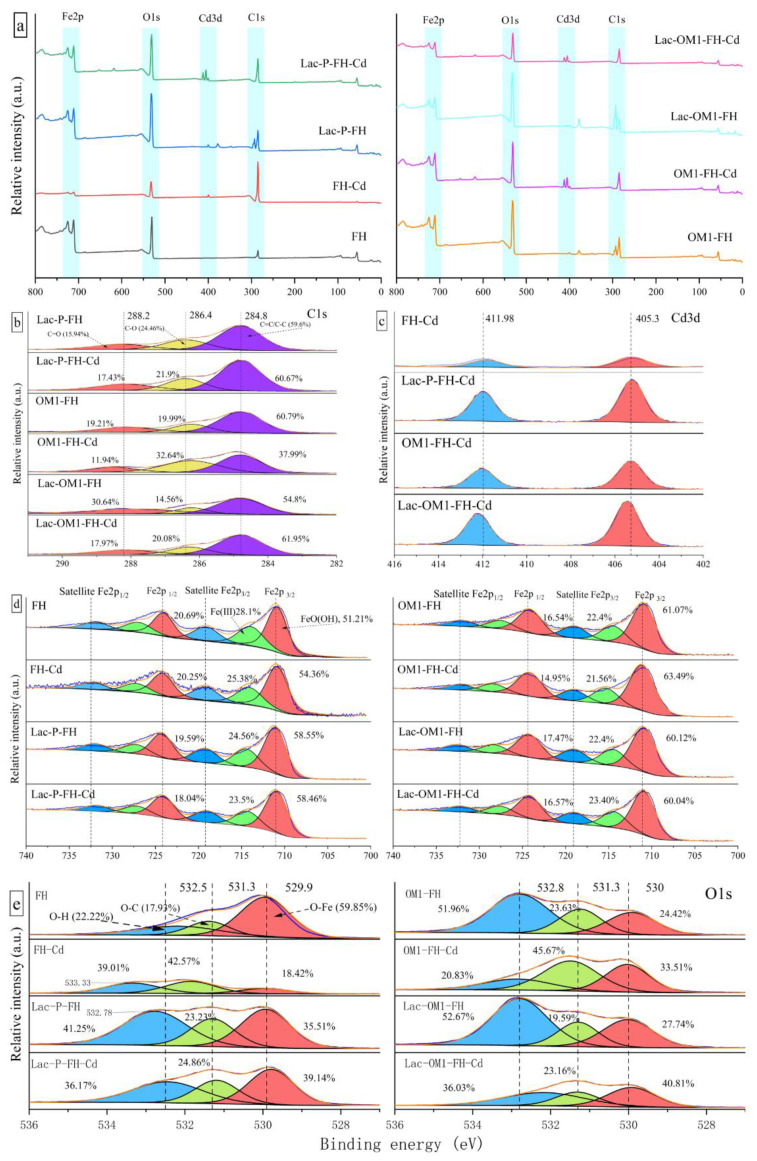
XPS spectrum of FH and Fe-OM before and after adsorption: (**a**) full XPS spectrum; (**b**) C1s spectrum of Lac-P-FH, OM1-FH and Lac-OM1-FH before and after adsorption; (**c**) Cd3d spectrum of FH, Lac-P-FH, OM1-FH and Lac-OM1-FH; (**d**,**e**) Fe2p and O1s spectrum of FH and Fe-OM before and after adsorption.

**Figure 10 ijerph-19-15650-f010:**
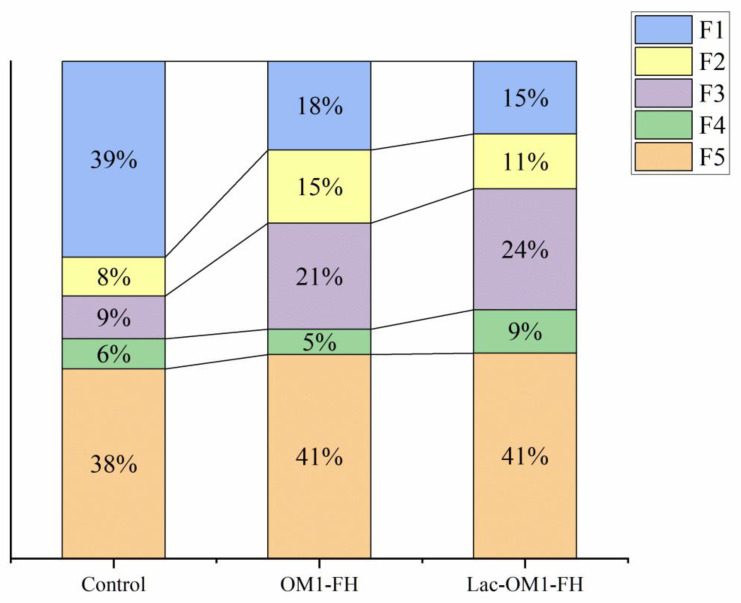
The variation of Tessier Cd fractions in soil sample. F1, the exchangeable Cd fraction; F2, the carbonate-bound Cd fraction; F3, the iron–manganese oxide-bound Cd fraction; F4, the organic matter-bound Cd fraction; F5, the residual Cd fraction.

**Table 1 ijerph-19-15650-t001:** Fitting data of adsorption kinetics and isotherms adsorption of Cd onto FH and Fe-OM associations.

Adsorption Kinetics													
	Elovich Q_t_ = a + blnt			Pseudo First-Order In(1 − Qt/Qe) = −kt		Pseudo Second-Order t/Qt = 1/(KQe^2^) + t/Qe		Two-Constant In Qt = a + blnt			Intra-Particle Diffusion Qt = K t^1/2^		
	a	b	R^2^	K	R^2^	Qe	R^2^	a	b	R^2^	K	b	R^2^
FH	0.5139	7.2054	0.933	0.0075	0.795	10.1523	0.9999	0.0568	1.9954	0.9187	1.015	8.3154	0.7238
LAC-P-FH	1.8244	9.8186	0.9919	0.0065	0.9421	21.367	0.9978	0.1082	2.3961	0.9889	0.3931	13.458	0.9156
OM1-FH	0.9794	9.0347	0.974	0.006	0.9631	15.36	0.9986	0.0759	2.522	0.9756	0.2147	10.954	0.931
OM5-FH	0.5645	8.9008	0.9228	0.0059	0.7382	12.195	0.9997	0.0516	2.2031	0.9114	0.1123	10.113	0.7261
LAC-OM1-FH	1.7978	6.9645	0.9938	0.0074	0.9727	18.083	0.999	0.1304	2.1122	0.9819	0.3819	10.601	0.8915
LAC-OM5-FH	0.6822	9.5116	0.9734	0.0061	0.8718	13.624	0.9996	0.0565	2.2755	0.9642	0.1406	10.931	0.8223
Isotherm adsorption													
	Langmuir equation Ce/Qe = 1/KQ_m_ + Ce/Q_m_						Freundlich equation InQe = In k + nIn Ce						
	Q_m_ (mg/g)		K		R^2^		n		In k		R^2^		
FH	39.53		0.0403		0.9861		0.5046		1.1383		0.9772		
LAC-P-FH	117.65		0.0343		0.9804		0.6635		1.6706		0.9627		
OM1-FH	86.96		0.0240		0.9809		0.6802		1.1		0.981		
OM5-FH	59.52		0.0183		0.9845		0.6864		0.5321		0.9825		
LAC-OM1-FH	100.00		0.0208		0.9753		0.7028		1.0872		0.9885		
LAC-OM5-FH	68.49		0.0163		0.9809		0.709		0.5306		0.9848		
LAC-P	68.97		0.0275		0.9887		0.6471		1.046		0.947		

t (min): adsorption time; Q_t_ (mg/g): the amount of Cd^2+^ adsorbed at time t; Qe (mg/g); the amount of Cd^2+^ adsorbed at equilibrium; a, b, K: adsorption kinetic constants. Ce (mg/L): concentration of Cd^2+^ remaining at equilibrium; Qe (mg/g): amount of Cd^2+^ adsorbed at equilibrium; Q_m_ (mg/g): theoretical maximum adsorption amount.

## Data Availability

Not applicable.

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
