# Peer review of "Enhanced Adsorption of Cd on Iron–Organic Associations Formed by Laccase-Mediated Modification: Implications for the Immobilization of Cadmium in Paddy Soil"

_ijerph, 2022, doi:10.3390/ijerph192315650_

Round 1

Reviewer 1 Report

The reviewer greatly thanks the Editor for having allowed reviewing this manuscript. The review paper ijerph-2028477 entitled " Enhanced adsorption of Cd on iron-organic associations formed by laccase-mediated modification: Implications for the immobilization of cadmium in paddy soil" is attractive and has information relevant to the audience. The paper is generally well-written and structured. However, some revisions must be conducted before its acceptance. Here are some general comments:

1.     The authors should include the work's objectives in the abstract before explaining the experimental design.

2.     In lines 63-64, the authors mention that many studies have confirmed that newly-formed amorphous iron (hydro) oxides preferentially bind phenolics and aromatic organic matter, but they only provide two references, the same problem from lines 78-81. I suggest they give at least four, which must be very recent.

3.     The authors should explain the paper's novelty in the introduction before stating the work's objectives.

4.     Line 101, adding the geographic coordinates would sound better.

5.     The lab instruments (names, selling company's name, city and country) used for lab analyses should be provided.

6.     In the "results and discussion" section, in most cases, the authors only state the results without giving the scientific reason behind the differences for comparison. I suggest that for every significant difference, the cause should be provided.

7.     There are some grammatical errors and punctuation that must be fixed and polished.

Reviewer 2 Report

The paper of  Weilin Yang, reported the evaluate the cadmium adsorption capacity of iron-organic associations (Fe-OM) formed by laccase-mediated modification, and assess the effect of Fe-OM on the immobilization of cadmium in paddy soil. The work is interested and authors presented comprehensive set of experimental results which is important for researcher’s community working in this field and applications. Yet, the work art need to be more observable, in particular labels. Also, Fig.5 (A,B) there are Chinese characters need to be translated. Other than tis, I recommend this manuscript to be published.

Reviewer 3 Report

Reviewer

MDPI – Int. J. Environ. Res. Public Health

Manuscript Number: ijerph-2028477

Title: « Enhanced adsorption of Cd on iron-organic associations formed by laccase-mediated modification: Implications for the immobilization of cadmium in paddy soil».

.

line 104 What units of measurement should be indicated here: % or mg / kg?

line 172 Statistical Analysis should be in the end of Materials and methods.

line 173-187 Each Equation should be numbered: Equation 1 ... .11.

Figure 1. Where is the statistical data processing? What is the difference between the options?
